# Collective Rescue: Why the Government Can Collaborate with the Public during Extreme Precipitation

Shupeng Lyu, Gongcheng Gao ⬤, Ching-Hung Lee *⬤, Lishuang Sun, Ning Xu and Chen Qian

School of Public Policy and Administration, Xi'an Jiaotong University, Xi'an 710049, China;
shupenglv@126.com (S.L.); vouslg@stu.xjtu.edu.cn (G.G.); sdusls@126.com (L.S.); doris_rio@163.com (N.X.);
qian_chen@stu.xjtu.edu.cn (C.Q.)
* Correspondence: leechinghung@xjtu.edu.cn

**Abstract:** Extreme precipitation poses significant threats to economic development and social security. In order to tackle this formidable issue, the Chinese government has invested substantial investments to promote the "sponge city" project. However, extreme precipitation in Zhengzhou on 21 July 2021 proved that this effort fell far short of its goal, highlighting the difficulty in relying solely on the government to respond to severe disasters. Collaborations between the public and the government have become essential for effectively managing extreme precipitation risk. However, bridging the gap between the public and the government remains a challenging task in China. Interestingly, an Online Collaborative Document (OCD) emerged without any financial investment from the government yet went on to save countless lives. The key lies in that the OCD greatly promoted the cooperation between spontaneous volunteering and official response. This study aims to explain how an OCD encourages effective collaboration between spontaneous volunteering and official responses in tackling extreme precipitation risk. This study employs the method of the case study about the extreme precipitation and analyzes the directed survey on the spontaneous volunteers, grassroots government officials, and affected individuals who used the OCD. Additionally, a variety of data collection techniques, including focus groups and one-on-one interviews, are used to examine the relevant information. This study explores how the OCD eliminated the paradox of collaboration between spontaneous volunteering and the official response thereby fostering coordination between them.

**Keywords:** spontaneous volunteering; official response; collaboration; digital platform; extreme precipitation

## 1. Introduction

An extreme precipitation occurred in Zhengzhou, the capital of one of the most populous provinces in China on 21 July 2021. Water levels rose rapidly within hours, destroying most of the public infrastructure including power lines, roads, communications, and sanitation facilities. The disaster caused a total of 14.786 million people to be affected, including 380 deaths and missing people with a direct economic loss of 40.9 billion RMB in Zhengzhou. The losses were so heavy that it even caught the attention of the top leaders of the Communist Party of China (CPC). In response, the General Office of the State Council promptly dispatched a specialized investigation team to conduct in-depth investigations, a signal in China that this disaster was being taken very seriously at the highest political level. There are two ironies in this disaster: on the one hand, Zhengzhou had just completed an investment of more than 50 billion RMB in a "sponge city" project. As the name suggests, a sponge city is designed to make the city we live in like a sponge that can store water when it rains by optimizing drainage capacity, the goal of which was precisely to deal with urban flooding caused by extreme precipitation. Since 2016, Zhengzhou has built a drainage network of over 5000 km, which was reported to absorb 70% of precipitation. However, the results of the disaster seemed to demonstrate that this investment had not proven

successful. On the other hand, a life-saving document, aptly named, emerged without any financial investment from the government yet it went on to save countless lives.

Disasters driven by water risks have been occurring with increasing frequency worldwide as the global climate continues to change, posing a serious threat to human survival [1,2]. During the 20-year period from 1998 to 2017, climate-related and geophysical disasters caused 1.3 million deaths and affected 4.4 billion people, leaving them injured, homeless, or in need of emergency assistance. Meteorological disasters, specifically, caused an average of 4066 deaths per year from 1984 to 2014 [3]. Above all, the extreme precipitation in Zhengzhou is not the first disaster related to water risks. Worse still, such disasters are unlikely to be the last. As cities expand, the uncertainty of water risks is rapidly increasing. Disasters related to water risks have become a major concern, and relying solely on official response is no longer adequate in effectively addressing these increasingly complex situations [4]. Consequently, collaboration between governments [5,6], and coordination, coproduction, and cocreating between the public, non-governmental organizations, businesses, and the government in response to disasters [7] have garnered significant attention particularly in the context of the New Normal and the digital transformation era that we are currently facing. However, the role of the public, a critical component of social forces, has long been overlooked in disasters [5,8]. In reality, during most natural disasters, the first responders are predominantly the public who utilize mobile social media platforms [9]. Compared with the government, the public who are undergoing natural disasters have stronger advantages when using advanced digital platforms for effectively and quickly taking actions, especially when facing public disasters [10,11]. Public participation and crowd sourcing can also enhance social disaster resilience in this New Normal and digital transformation era [9,12]. Hence, it is essential to incorporate the power of the public into official emergency responses.

Citizen volunteers have utilized spontaneous volunteering as a crucial means of engaging in emergency response efforts. A notable example occurred in 2011 when a devastating tornado struck Goderich, Canada, prompting over 7000 people to join a Facebook group called "Goderich Ontario Tornado Victims and Support". Through community-wide information sharing, thousands of affected individuals were able to receive the necessary assistance [13]. Similarly, during the 2015 European refugee crisis, numerous citizens participated in emergency rescue efforts spontaneously, often stepping in to complement official response systems [14]. Despite all this, incorporating spontaneous volunteering into official responses remains challenging [15]. One of the main reasons is that official agencies frequently grapple with coordination difficulties when it comes to integrating unplanned spontaneous volunteers [16].

In recent years, the advancement of internet technology has fostered the growth of various digital platforms, offering potential solutions for integrating spontaneous volunteering into official response efforts [17,18]. Through digital channels, governments can establish easier communication with emerging volunteers, enabling the quick delivery of relevant information to official agencies. This seamless information transfer enhances effectiveness and creates opportunities for collaboration between the two parties [19,20].

The collaboration between spontaneous volunteering and official response has always been a focal point, while digital platforms continue to reshape the emergency response process [5]. However, research into the role of digital platforms in the collaboration between the two parties has been relatively scattered, with a primary focus on social media as the subject of investigation. Yet, with the ongoing advancement of digital technology, innovative digital platforms continue to emerge. Following the outbreak of COVID-19, online collaborative office platforms gradually emerged in the New Normal and digital transformation era; these tools not only played an important role in work but also played an unexpected role in emergency responses to natural disasters. During the extreme precipitation in Zhengzhou, China in July 2021, an Online Collaborative Document (OCD) successfully harnessed the power of spontaneous volunteers and official agencies, achieving an efficient emergency response. OCDs are an efficient tool that apply a multi-person

collaborative authoring model to text editing. The technology is similar to Wikipedia in terms of stimulating the potential for group intelligence. It is worth noting that an unexpected OCD, surprisingly, saved many people's lives at a moment when the sponge city was paralyzed, which no one realized could be done. Anyway, no matter how reliable the flood defenses are, it is difficult for them to withstand an hourly precipitation that reaches 201.9 mm. Therefore, it is worth exploring whether digital platforms, such as OCDs, are a potential way to deal with such serious disasters.

In fact, during the initial stages of the extreme precipitation, unorganized spontaneous volunteers widely disseminated information about the disaster on Weibo (China's largest social network, comparable to Facebook), while the government barely acknowledged this information. Hence, this research aims to answer research questions (RQs) about how to make collaboration between spontaneous volunteering and official response. RQ1: What are the fundamental conditions for collaboration between spontaneous volunteering and official response? RQ2: How did the OCD integrate spontaneous volunteering into official response? Evaluating the impact of the OCD on the collaboration between spontaneous volunteering and official response systems is complex and can be observed in specific crisis scenarios where quantitative analysis methods may not be applicable. Hence, we have adopted a case study approach to thoroughly examine this issue, providing profound insights into the case itself and gathering a substantial amount of analytical information that reflects the process of occurrence, development, and change.

This paper begins with a brief review of spontaneous volunteering during natural disasters and the impact of digital platforms on spontaneous volunteering in Section 2. Section 3 provides an analytical framework followed by information about the case and the research process. Then, a case study is illustrated based on the analytical framework in Section 4. Finally, a conclusion and discussion are presented on how official responses should promote the positive role of spontaneous volunteering in emergency responses through digital platforms.

## 2. Literature Review

### 2.1. Spontaneous Volunteering and Official Response in Disasters

Spontaneous volunteering refers to volunteer activities that occur suddenly and spontaneously after a disaster, which are not part of the official response plan. Such activities are characterized by a chaotic situation and a large influx of volunteers in a short period of time [21,22]. Research on spontaneous volunteering has focused on its contributions, the challenges it poses to official response, and solutions for integrating spontaneous volunteering. One advantage of spontaneous volunteers is that their help is easy to obtain and comes at a low cost [23]. Additionally, since the spontaneous volunteers may be in the affected area, they are able to learn about the situation of disaster victims and react more quickly than official agencies [24]. The characteristics that come from being in the disaster area make spontaneous volunteering advantageous in searching for information [15]. However, spontaneous volunteering has also posed many challenges for the official response. A surge in spontaneous volunteers has intensified chaotic situations and hindered official responses. At the same time, spontaneous volunteers usually have not received professional training and are thus exposed to huge risks during disaster events [25]. Overall, spontaneous volunteering is in an involvement/exclusion paradox [22]. On the one hand, official agencies require more personnel and resources in disasters [26]. On the other hand, they have many reasons to refuse the participation of spontaneous volunteers. This means that how to scientifically manage large-scale spontaneous volunteering to achieve better response effects is a topic worthy of attention.

Existing research has extensively explored this issue from the official response's perspective. Firstly, spontaneous volunteering has a completely different structure from official response. Official agencies are often restricted by top-down, command, and control frameworks, which leaves them with few alternative measures. In contrast, spontaneous volunteering is unstable. The temporary organizations formed by spontaneous volunteers

lack leadership, and their members and goals are constantly changing over time. The team may be formed or disbanded at any time; compared with official response, spontaneous volunteering has a very high degree of uncertainty [27]. Therefore, the management methods of official agencies are completely unsuitable for managing spontaneous volunteers [15]. Some studies suggest that in the face of spontaneous volunteering, official agencies should play the role of "shepherd" to guide public participation, fully leveraging the role of spontaneous volunteering, rather than limiting spontaneous volunteers' work content and responsibilities [28]. At the same time, official agencies can strengthen emergency drills in daily life, so that spontaneous volunteers can respond when disaster strikes [29,30]. In addition, some studies point out that official agencies can strengthen community capacity through establishing close links within the community, which also helps spontaneous volunteering play a role in disaster events [31]. However, the implementation of these solutions is based on the premise that official agencies view spontaneous volunteering as a potential emergency response resource before a disaster occurs. In fact, the implementation of these measures itself means complex coordination. Therefore, even if official agencies must collaborate with spontaneous volunteers, they are more inclined to form collaboration with the public through crowd sourcing rather than through having intimate relationships with the public [32]. It can be said that the elimination of obstacles to collaboration between official response and spontaneous volunteering in disasters is still an urgent problem that needs to be solved.

### 2.2. The Role of Digital Platforms on Spontaneous Volunteering

The development of digital platforms provides new opportunities for spontaneous volunteering. In summary, digital platforms play two roles: one is to provide a channel for integrating resources, and the other is to enhance the ability of spontaneous volunteering to participate in emergency response. These two roles are closely related, and research on the role of digital platforms in spontaneous volunteering analysis is mainly based on these two aspects.

Firstly, digital platforms accelerate the speed and expand the scope of information dissemination, enabling people to quickly obtain relevant information about disasters [33]. Meanwhile, the problems faced by spontaneous volunteering become more complex. A previous study has pointed out that spontaneous volunteering using digital platforms is similar to a Garbage Can model, where participation is fluid, preferences are different, and technical means are unknown [34]. Nonetheless, what is different is that the public is gradually moving from disorder to order due to the emergence of digital platforms [35].

This effect may be due to the fact that digital platforms can gather spontaneous volunteering from different times and spaces, which is convenient for information dissemination and communication [36]. In the past decades, official agencies have not been clear about how to communicate with spontaneous volunteers. This issue has been partially addressed with the existence of digital platforms. Through the help of digital platforms, even in the face of major crises, spontaneous volunteering will not fall into an anarchic response state [37]. Secondly, spontaneous volunteering is basically maintained in social networks, and even completely unfamiliar people can establish contact through digital platforms [38]. Based on the network provided by digital platforms, spontaneous volunteers are more easily formed into a distributed collaborative organizational structure, which enhances the agility of its actions and the stability of its structure [39].

Overall, digital platforms provide a channel for the orderly organization of spontaneous volunteering, but existing research only proposes suggestions for the role of digital platforms, and few studies have conducted comprehensive analysis based on the role of digital platforms in disaster events. This paper will explore the conditions of the occurrence and process of digital platforms in integrating spontaneous volunteering into official response processes based on the life-saving document that occurred after the extreme precipitation in Zhengzhou, China in 2021, and provide theoretical and practical value for

better utilizing digital platforms to fully leverage the effects of spontaneous volunteering in disasters.

## 3. Theoretical Framework

### 3.1. Process of Digital Platforms Integrate Spontaneous Volunteering into Formal Response

The theory of coproduction [40] points out three necessary conditions for collaboration between the government and the public: (1) the public should play a role in compensating rather than replacing the government's functions. If the public can replace the government in providing public services, then there is no value in collaboration between the two parties; (2) The cost of the public provision of services should be lower than the cost of the government's. If the cost of the public provision of services is higher than that of the government, then the government should still provide public services solely; (3) The government is subjectively willing to collaborate with the public; otherwise, even if the public can leverage their own advantages to fill the gap in government functions, the government may still refuse to cooperate with the public for various reasons (legitimacy, fairness, accountability, etc.). The coproduction issues that Ostrom is concerned about mainly occur in real space; however, crisis events greatly reduce the possibility of coproduction in real space. Although the public has access to a large amount of emergency information needed by the government in crisis situations, they often face a huge collaboration gap due to the physical distance between them.

Since collaboration between the public and the government in crisis situations is difficult to occur in real space, digital platforms help diverse entities achieve linkage, interaction, matching, and value cocreation. The emergence of innovative digital technologies provides an opportunity to shift the collaboration of the government and the public from real to virtual space. This phenomenon is well summarized by the disembedding theory. Disembedding is a concept proposed by British sociologist, Anthony Giddens, which refers to the extraction of social relationships from regional relationships that interact face-to-face with each other. This includes both extracting social relationships from regional relationships and reconstructing them over time [41]. The disembedding of multiple elements in the specific spatiotemporal context also provides opportunities for these elements to form new systems. It can also be said that the essence of disembedding is the reproduction of social structure, functions, and actions. The disembedding theory explains the process of public and government collaboration from real to virtual space, and the coproduction theory explains the conditions of their collaboration. Therefore, this study proposes a disembedding and reproduction framework from the three dimensions of "structure → power → action" to explain the conditions of collaboration between spontaneous volunteering and official response in disasters (As shown in Figure 1).

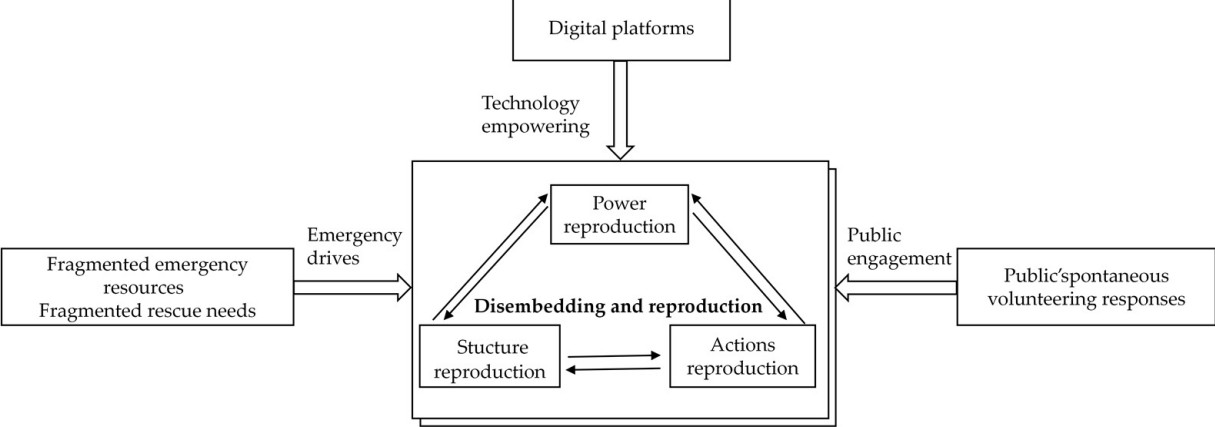

**Figure 1.** Theoretical Framework.

The first is the structure reproduction. Clear responsibilities and a strict division of labor are important features of the bureaucratic system and the foundation for ensuring administrative efficiency. Spontaneous volunteers are in a fragmented and unorganized state when disaster strikes, and this difference in organizational structure often leads to a chaos of public participation. The digital platform pulls the spontaneous volunteers and official agencies out of their physical space and constructs new structures in a new space. Specifically, digital platforms provide the public with the opportunity to freely create without worrying about the awkwardness of communication with strangers and the pressure of engagement caused by accountability [42]. Similarly, for official agencies, communication based on digital platforms can also reduce the institutional costs brought about by bureaucratic systems [43].

The second is the power reproduction. Power is not only the foundation of effective action, but also the source of ineffective action. The top-down control system of the bureaucratic system ensures the effectiveness of personnel actions within the system, but it lacks the ability to supervise public behavior, or in other words, supervising public behavior has extremely high costs. The digital platform provides an effective tool for official agencies to monitor the behavior of spontaneous volunteers. The public on a digital platforms can engage in self-regulation and behavioral correction, and research has shown that such self-regulation is effective [39]. This means that the cost of supervision by the official agencies for the spontaneous volunteers is transferred to within the spontaneous volunteers, reducing the cost of integrating spontaneous volunteering into the official response.

The third is the actions' reproduction. Public participation has been marginalized in official responses for a long time. Even if spontaneous volunteers are helpful to official agencies, it is still possible for official agencies to refuse to collaborate with the spontaneous volunteers due to the lack of collaboration actions in both parties' actions [44]. On digital platforms, information constitutes the actions of collaboration between both parties, and spontaneous volunteers become an information hub for official agencies and the affected; where the public organizes and transmits distress messages, the official agencies obtain this information through digital platforms [12].

The above three elements interact and promote each other, shaping the collaboration between the spontaneous volunteers and the official agencies. In practice, disembedding and reproduction is aimed at bridging the fragmentation of emergency resource supply and demand. Spontaneous volunteers are the initiators of disembedding and reproduction and, together with official agencies, form the main actors of disembedding and reproduction. The development of digital technology has played an important role in enabling this process. Based on the empirical case of the OCD used following the extreme precipitation in Zhengzhou, China in 2021, this paper will deeply analyze the process of a digital platform promoting the collaboration of spontaneous volunteers and official agencies, with a view to providing useful thinking for better integrating public forces and coping with the impact of water risks.

### 3.2. Methodology

This is a qualitative case study. Since the focus of the study is exploratory, the role of digital platforms is closely related to an external scenario, and intervention cannot be made on the focal phenomenon. Therefore, a case study was selected, which had typical characteristics and can concentrate on reflecting important features of a category phenomenon in sociological research. Rich descriptions can provide more details for the research. Through a single case study, the phenomenon can be deeply explored, which can highlight the static phenomenon scene and systematically demonstrate the causal mechanism and process [45].

The life-saving document, born during the Zhengzhou extreme precipitation, marked the emergence of the Online Collaborative Document (OCD) as an officially recognized platform with extensive participation in dealing with natural disasters. Subsequent extreme precipitation events in provinces like Shanxi and Guangdong in China witnessed the

reemergence of similar OCDs, further facilitating collaboration between spontaneous volunteers and official agencies. Although these subsequent OCDs largely exhibit similar characteristics to the life-saving document, the original serves as a typical case that reflects the main features of OCDs, thus contributing to the external validity of our study.

## 4. Case Study of Life-Saving Document of the Zhengzhou Extreme Precipitation

The continuous precipitation occurred in Zhengzhou, Henan, China from 18 July 2021, 18:00 to 21 July 2021, 00:00, with an accumulated average precipitation of 449 mm, which resulted in extremely severe urban waterlogging. In the extreme precipitation, any corner of the city could become a vulnerable point with numerous and scattered affected people. The official agencies were overwhelmed, and many people's cries for help were difficult to transmit to them. Both inside and outside the disaster area, spontaneous volunteers hoped to help the affected people as soon as possible in their own ways. However, in offline scenarios, the official response was already stretched thin, and lacked the ability to integrate spontaneous volunteering. In online scenarios, the internet information was severely overloaded, and it was also challenging for official agencies to identify effective information.

At 20:57 on 21 July, an OCD called Information of People Awaiting Rescue (later known as the life-saving document) appeared on the internet. Initially, the document was maintained by only 30 people, and its main function was for volunteers to collect information on people in need and rescuers on Weibo, and to match them up. However, as time went on, more and more people saw this document and started to fill in information such as "trapped in the subway", "pregnant women about to give birth", and "lacking food and water". While distress messages emerged, numerous compassionate individuals also contributed valuable information such as "medical guidelines", "safe havens", and "dangerous areas" to the document. Unlike Weibo, a large amount of information did not lead to disorganization in the OCD. Meanwhile, netizens began to spontaneously classify and sort document information, such as assigning priority to urgent help-seeking requests. Everyone could edit the document without causing chaos; instead, the functionality of the documents was continuously optimized. Everyone had the right to edit and modify the document, and was fully capable without succumbing to technical barriers. Thus, people from all walks of life across the country participated in the editing of the life-saving document, whether they were doctors, programmers, retired soldiers with a certain skill, or ordinary students and working people. According to the post-disaster statistics, nearly 300,000 people participated in editing during the Zhengzhou extreme precipitation, and about 3000 people were rescued.

It is worth noting that since the creation of the life-saving document on 21 July, the content of the document continued to increase, gradually attracting the attention of official departments. Starting from 22 July, the Zhengzhou grassroots government collected information from the document and responded to the distress messages uploaded by spontaneous volunteers on the OCD. The requests for help in the OCD came from different regions, and the grassroots governments communicated with each other in real-time to achieve the timely processing of the distress messages in the document, greatly improving the speed of obtaining distress messages and the efficiency of emergency rescue by the official agencies. In the Zhengzhou extreme precipitation, spontaneous volunteering based on digital platforms provided information to official departments, and the grassroots government promptly provided rescue to the trapped masses; spontaneous volunteering did not cause chaos. On the contrary, it was well integrated into the official response. The process of the life-saving document exerting its effect is shown in Figure 2.

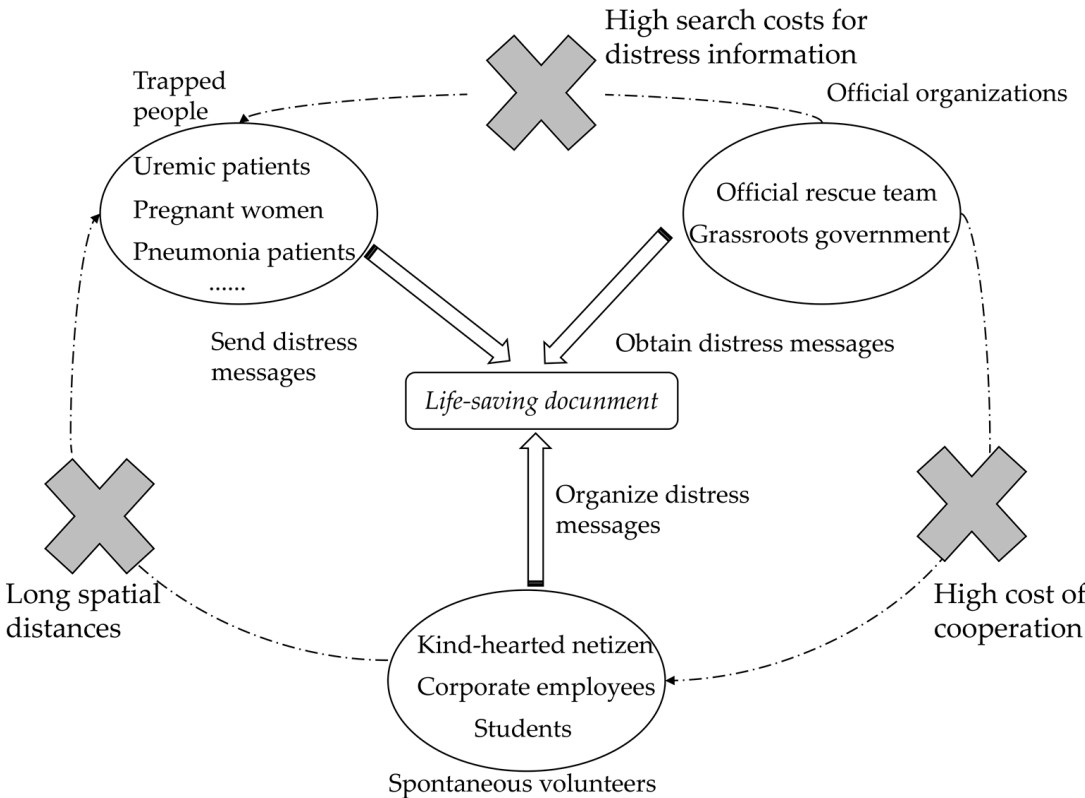

**Figure 2.** The process of life-saving document exerting its effect.

### 4.1. Data Collection and Data Descriptions

The materials used in this analysis are a combination of primary and secondary sources. Primary sources refer to practical data collected through in-depth interviews, the basic information of which is shown in Table 1. Between June and July 2022, we conducted interviews with victims of the Zhengzhou extreme precipitation, members of the life-saving document team, and grassroots government officials who participated in the official response, in order to gain insight into the interaction difficulties between the official response and the spontaneous volunteering and the creation process and role of the life-saving document, and to accumulate more than 50,000 words of primary data. Secondary sources include news reports from China Central Television (CCTV) and People's Daily on the life-saving document, Tencent's special report: 24 Hours of a life-Saving document, and the final online form of the life-saving document. These materials enrich the evidence presented in this paper and improve the reliability of the case analysis results. They also serve as a means of validating the effectiveness of the interview data. The information of the interviews is shown in Table 1.

**Table 1.** Information of interviews.

| Interview Code | Date | Interviewee | Method |
| --- | --- | --- | --- |
| SZ01-20220625 | June 2022 | Citizen affected by the Zhengzhou extreme precipitation | Focus group |
| JM01-20220701 | July 2022 | Member of the operation team for life-saving document | Many-on-one interview |
| ZF01-20220703 | July 2022 | Grassroots staff who use the life-saving document for emergency rescue | Many-on-one interview |
| ZF02-20220715 | July 2022 | Organizational personnel of government | One-on-one interview |

*4.2. Process of Digital Platforms Integrate Spontaneous Volunteering into Official Response*

4.2.1. Structural Reproduction: Activation of Collaborative Networks

During the rainstorm, a large number of spontaneous volunteers formed a social network, which is characterized by many nodes and loose connections. The official agencies form a hierarchical network, characterized by fewer nodes and closer connections. In a social network, information overload in social media makes it difficult for the nodes in the network to interact in a timely manner, and numerous pieces of information are "submerged" in the flood of information, making it difficult to carry out the massive rescue needs. In a hierarchical network, due to the size limitations of the network, it is difficult to integrate a large number of spontaneous volunteers into the network.

*"The information on those two days of the rainstorm was too dense. I posted a distress message for my friend on my Weibo, but it was quickly lost once it was refreshed. There was no way. If you didn't have many fans or popularity, your information would be difficult to get attention in that case."* (SZ01-20220625)

*"There were many people who wanted to volunteer at our street office, and we had to reject about fifty or sixty people a day. These volunteers are not part of our staff, and we needed to consider their travel costs, food and personal safety, so we cannot let them come."* (ZF01-20220703)

A life-saving document activates the decentralized collaborative network between spontaneous volunteers and official agencies by integrating them. Firstly, life-saving documents create an emergency message space. During the extreme precipitation in Zhengzhou, the rescue information from different channels, such as posts and comments on Weibo, as well as updates on WeChat Moments, was summarized in the same OCD, which greatly improved the efficiency of official agencies in obtaining them.

*"We seldom use this kind of OCD in our work at ordinary times. We feel the role of OCD for the first time in this rainstorm, and we must admit that it (referring to" lifesaving document ") is really fast, much faster than the way of getting rescue information from phone calls."* (ZF01-20220703)

Secondly, a decentralized collaborative model promotes the stable and sustainable network. The public participation in emergency response behavior at different times and in different spaces can be integrated into one OCD. The decentralized collaborative network ensures that the number of volunteers is not limited by the traditional organizational scale, and the openness of the organizational structure also allows more public participation in the joint production process. For the government, the life-saving document also forms a temporary horizontal connection within the government, and grassroots governments independently divide their work based on the distress information in the life-saving document, forming a collaborative relationship.

*"The distress messages on the document belongs to different jurisdictions, and the sub tables of the document also belong to different departmental functional areas. We can communicate information between different units through this document. For example, if we see relevant information belonging to other units in the document, we will forward it to them and tell them which row and column in the table contain the information they need."* (ZF01-20220703)

4.2.2. Power Reproduction: Relocation of Governance

Official responses face a challenge in engaging with spontaneous volunteering in how to ensure that spontaneous volunteering during emergencies align with the official departments' expectations during emergencies. This requires coordination costs. In disaster situations, institutionalized approaches have limited effectiveness in mobilizing public participation.

*"It's unrealistic to recruit volunteers for urgent tasks like carrying sandbags or rescuing people, as it's impossible to gather so many people in a short period of time. In fact,*

*we had very few volunteers during the extreme precipitation in Zhengzhou because of a bad experience in recruiting volunteers during the COVID-19. Some people become unruly when given even a little power, such as maintaining the order of the testing queue. They would allow their friends and relatives to cut in line, and we had no choice but to assign one of our staff members to each volunteer position to ensure that they only did supporting work. However, we found that this would consume additional manpower. So, we rarely recruit volunteers now."* (ZF02-20220715)

The approach adopted by the life-saving document in response to this issue was to empower every member of the volunteers with the right to make judgments. The decentralized and collaborative nature of the OCD endowed individuals with the power to reject improper information. For the government, this means that the cost of overseeing spontaneous volunteers was transferred to every participant in the creation and editing of the life-saving document.

*"Sometimes the information provided by the volunteers seeking help is not entirely accurate. To address this, we have a team dedicated to verifying the accuracy of the information before we contact rescue teams for assistance. Unfortunately, there are also some individuals who intentionally sabotage our efforts by deleting the document. To prevent this, we have volunteers who regularly back up the document to ensure its smooth operation."* (JM01-20220701)

4.2.3. Actions Reproduction: Emergency Driving

One issue requiring further clarification is the rationale behind official agencies responding to spontaneous volunteering despite no cooperative costs between the government and the public. Two reasons account for this. Firstly, spontaneous volunteering gains lots of distress messages from various channels and aggregates them in the document. As the number of requests grows, official agencies need to respond to them to maintain legitimacy. Secondly, a life-saving document is an unfamiliar tool for official departments, and with limited time resources, they must tread carefully before using one. There exists a time lag between the response of spontaneous volunteering and that of the official agencies, and the OCD boasts a high degree of openness. Both official agencies and spontaneous volunteers have ample space for choices upon viewing the document, perceiving pressure, engaging cautiously, and responding gradually. The actions of spontaneous volunteering and the official response are coupled in this process.

*"When the disaster happened, we had to act fast because there wasn't much time to waste. With so many requests for help coming in, we needed to make sure everyone was safe and avoid any further harm. The first thing we were worried about was whether these requests for help were genuine or not. Once we confirmed the accuracy of the information, we had people monitoring the document constantly. Whenever there was a new request within our jurisdiction, we would contact the relevant department for rescue operations. If it was outside of our jurisdiction, we would notify the other local authorities to handle it. It's just like how earthquakes and heavy rains have a critical period of 72 h for rescue operations."* (ZF01-20220703)

In the collaboration process between the official response and spontaneous volunteering, both parties achieved behavioral coupling through information exchange. Spontaneous volunteers organized the information, and official agencies obtained the information and implemented rescue operations based on the OCD. This strategic choice enabled the collaboration between the official agencies and spontaneous volunteers to be ultimately realized in practical situations. This process is shown in Figure 3.

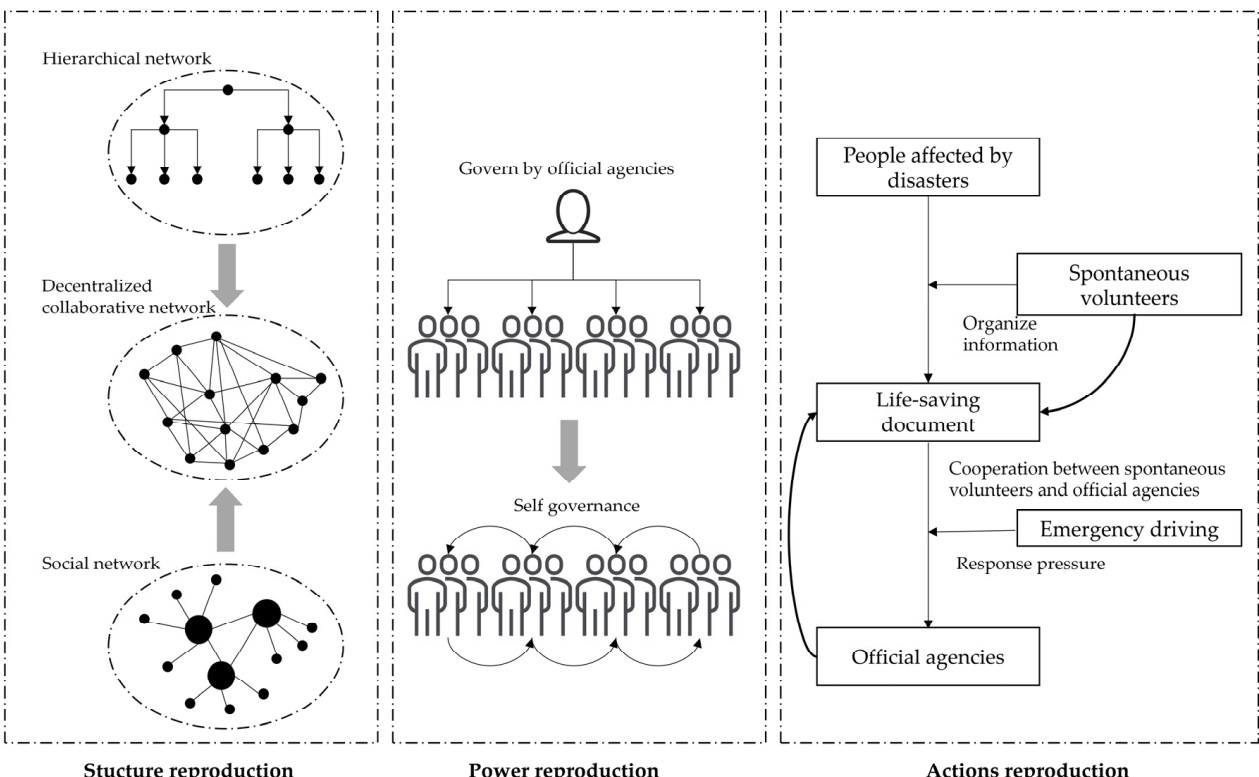

**Figure 3.** Process of disembedding and reproduction in life-saving document.

## 5. Discussion

The coproduction between spontaneous volunteering and official response is an important issue that has long been overlooked. Previous research has identified the characteristics of spontaneous volunteering and official response, as well as the difficulties faced by both parties in collaboration. It has also explained the effectiveness of digital platforms in improving public participation and government emergency response efficiency. However, existing research still lacks a clear understanding of the logic behind digital platforms, promoting collaboration between spontaneous volunteers and official agencies.

In this study a novel framework that explores the role of OCD in fostering collaboration between spontaneous volunteering and official responses during natural disaster events was proposed. Furthermore, through an in-depth case analysis, we attempts to provide a practical solution to address the intricate challenges associated with the collaborative dynamics between spontaneous volunteering and official responses. The result provide valuable insights into the role of new digital platforms in disaster events in the internet era. However, as a new form of collaboration, there are still some obstacles in the practice.

Firstly, the accessibility of emergency resources is hindered by the digital divide. In the case of extreme precipitation in Zhengzhou, China, most of the people who created and maintained the life-saving document were young and middle-aged people living in cities, and most of the people who uploaded the rescue information were students and white-collar workers. That is to say, elderly people with higher vulnerability to disasters and residents living in rural areas did not participate in this collective action. If the official agencies prioritize responding to the distress information in the documents, it may make the already vulnerable groups more vulnerable. The emergency rescue of official departments belongs to public resources, and because it guarantees the safety of the public's life, it requires higher levels of fairness. Therefore, the issue of the digital divide will always be an important factor in limiting the development of coproduction in digital platforms.

Secondly, data security concerns arise due to personal privacy issues. In general situations, personal information is related to personal privacy and dignity, but in crisis

situations, the public attributes of information begin to be reflected. Rescue information is expected to be seen by more people. The efficient transmission of information inevitably requires individuals to give up personal privacy rights temporarily. Disembedding and reproduction also erases the order and rules in the existing space. In a virtual space, data security may become a major risk issue, and the feeling of being monitored and spied on will give the public anxiety during the process of using digital platforms, limiting the engagement of spontaneous volunteers on digital platforms.

Thirdly, the development of digital platforms is constrained by bureaucratic systems' limitations. Digital platforms are characterized by a flat organization and rapid communication, which contrasts with the hierarchical structure and processes of government systems. Specifically, in terms of regulations, with the support of digital platforms, the bottom-up information connectivity has significantly improved, while the top-down process of obtaining information faces institutional obstacles, such as accountability and the strict inspection of data. In terms of structure, digital platforms provide technical tools to break departmental boundaries, but this tool still has limitations. In this case, the use of the OCD was limited within grassroots governments. The OCD only broke through the barriers of grassroots governments; higher-level governments were not involved in the collaboration with spontaneous volunteers.

Given the potential and obstacles of digital platforms, it is imperative to explore ways to leverage these platforms to enhance collaboration between spontaneous volunteering and official response when combatting water risks. Firstly, it is essential to leverage the strengths of spontaneous volunteers while avoiding their shortcomings. While the participation of spontaneous volunteers in on-site rescue during disasters is seen as a commendable act, it is evident that their lack of professional skills and the challenges in coordinating their actions limit their effectiveness in this area [46]. However, the greatest advantage of spontaneous volunteers lies in their proximity to the disaster's epicenter and their ability to tap into the social networks of those affected by the disaster, thereby gaining access to valuable information about vulnerable individuals [15]. This study reveals that OCDs further amplify this advantage of spontaneous volunteers and emphasize the need to fully utilize the OCD in future emergency responses. By doing so, spontaneous volunteers' access to information can be harnessed to improve rescue efficiency. Secondly, it is crucial to dispel stereotypes associated with spontaneous volunteers in disasters and recognize the potential of collective actions. Just as previous studies have recognized the important role of collective action by spontaneous volunteers in responding to refugee crises [47], this study finds that this role is equally effective in responding to extreme precipitation. What is more, with the support of the OCD, the pace of organized collective action was accelerated, and its scale was expanded. While voluntariness at the individual level is critical to examining the motivations and mechanisms by which individuals participate in responding to emergencies [48,49], overlooking the impact of collective actions on emergency responses can lead to an incomplete understanding of public participation. The life-saving document demonstrated that, given the appropriate environment, every segment of the public has the ability to participate in emergency response, and collective efforts can yield greater impact. Thirdly, this study encourages an enhancement in the laws and regulations pertaining to the utilization of digital platforms like OCDs for emergency responses. Although OCDs have demonstrated their potential in emergency response [50], there is still a significant degree of uncertainty regarding the voluntary participation of spontaneous volunteers in the OCD. Therefore, it is crucial to establish a robust management system and legal system for voluntary volunteers on digital platforms, ensuring their consistent and reliable engagement in emergency response activities. Meanwhile, it is essential to standardize the process of data acquisition, prioritize data privacy, maintain data security, and ensure data quality.

## 6. Conclusions

This study examines the role of the OCD in promoting collaboration between spontaneous volunteering and official responses. We propose a disembedding and reproduction

framework to illustrate this issue. Specifically, it identifies three fundamental conditions that shape the collaboration between these two entities including (1) structure reproduction; (2) power reproduction; and (3) actions reproduction. Meanwhile, we propose the implementation of the OCD to meet these conditions, i.e., the activation of collaborative networks, the relocation of governance, and emergency driving, thereby overcoming the paradox of collaboration between spontaneous volunteering and official response.

**Author Contributions:** Conceptualization, S.L.; methodology, S.L. and C.-H.L.; formal analysis, S.L. and G.G.; investigation, L.S. and N.X.; data curation, C.-H.L.; writing—original draft preparation, S.L. and G.G.; writing—review and editing, C.Q. and C.-H.L.; visualization, G.G. All authors have read and agreed to the published version of the manuscript.

**Funding:** This research was funded by National Social Science Fund of China (grant number: 21BZZ079), Innovation Capacity Support Project of Shaanxi Province (grant number: 2017KRM011), and National Natural Science Foundation of China (grant number: 72174168).

**Data Availability Statement:** The data presented in this study are available on request from the corresponding author. The data are not publicly available due to privacy reasons. The recorded interviews would possibly allow conclusions to be drawn about a person's identity.

**Conflicts of Interest:** The authors declare no conflict of interest.

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
