# Peer review of "Collective Rescue: Why the Government Can Collaborate with the Public during Extreme Precipitation"

_water, doi:10.3390/w15152848_

Round 1

Reviewer 1 Report

The paper submitted for review is highly relevant in terms of its information and application potential.

The abstract has been prepared in accordance with current good scientific practice in this area.

Introduction - The authors correctly introduce the reader from the general subject area to the specific subject of analysis. They indicate the significance of the research undertaken. However, the introduction should also briefly explain the methodological approach used.

Line 98 - 'it' should be capital letter ...

The literature review and theoretical framework have been developed with due care. The only doubt is raised by the wording "uOCDul" (line 270). What is this supposed to mean?

Chapter 4 - In the case study, I miss a thorough explanation of the mechanism of the life-saving document (OCD).

Line 357 - 'Life-saving' instead of 'life-saving'.

Discussion section - In my opinion, this is a very well developed section of the article.

Conclusions section should be included in the Discussion section and the name of the section should be Discussion and Conclusion.

In conclusion, the paper requires some minor revisions before being published.

Author Response

Dear Editors and Reviewers:

We sincerely appreciate all the detailed comments and suggestions made by the reviewers. We have reproduced your comments so they can be considered point by point with our responses. We have also numbered the comments to help us organize our responses. All comments are considered carefully and summarized as follows, and all the revisions are marked in Blue color in the original paper for reviewers’ readability. Per your communication, the revised manuscript is submitted as such and we thank you for considering our manuscript for publication. The main corrections in the paper and the responds to the reviewer’s comments are as following:

Reviewer 1

1.The authors correctly introduce the reader from the general subject area to the specific subject of analysis. They indicate the significance of the research undertaken. However, the introduction should also briefly explain the methodological approach used.

Response:

We are very grateful for your valuable instructive advice. We apologize for the lack of a description of the methodology in the previous manuscript, we have briefly described the research methodology and the reasons for using it in the Introduction in response to your comments. (Line 115-120)

2.Line 98 - 'it' should be capital letter ...

Response:

We apologize for this error and have corrected it. We have also checked the manuscript for errors of the same type and made modifications accordingly. (Line 102)

3.The literature review and theoretical framework have been developed with due care. The only doubt is raised by the wording "uOCDul" (line 270). What is this supposed to mean?

Response:

Thank you for carefully reviewing the manuscript. This is a spelling error, and we have made modifications to this issue. We have also checked the manuscript for errors of the same type and made modifications accordingly. (Line 280)

4.Chapter 4 - In the case study, I miss a thorough explanation of the mechanism of the life-saving document (OCD).

Response:

We are very grateful for your valuable instructive advice. In the previous manuscript, we did not provide a thorough explanation to the mechanism of "life-saving documents". In order to help readers understand this part clearly, we have revised the case study to describe the mechanism of  the "life-saving document" in more detail. (Line 313-331)

5.Line 357 - 'Life-saving' instead of 'life-saving'.

Response:

Thank you for carefully reviewing the manuscript. We have made modifications to this issue. We have also checked the manuscript for errors of the same type and made modifications accordingly. (Line 391)

6.Discussion section - In my opinion, this is a very well developed section of the article. Conclusions section should be included in the Discussion section and the name of the section should be Discussion and Conclusion.

Response:

Thank you for this valuable comment. We consulted the format of the articles published in the special issue and found that conclusions and discussion are usually separate rather than merged, so we chose to keep the original format.

Reviewer 2 Report

Interesting paper with valuable approaches to climate challenges. 

Suggestions for improvements: 

Maybe shortly explain what is "sponge city"

Explain all the abbreviations, even if some of them seem obvious.

References are not alphabetically written.

Line 360 explain which channels - give a few examples. 

From Table 1 or in general it is not clear the size of the sample - is it representative? 

Maybe I miss even more input from the actual answers from the interviews and their "interpretation". 

There are grammar mistakes in lines: 82, 98, 172, 180, 216, 270, 286, 335, 357, 359, 463. 

Author Response

Paper Title: Collective Rescue: Why the Government can Collaborate with the Public during Extreme Precipitation?

Authors: Shupeng Lyu, Gongcheng Gao, Ching-Hung Lee*, Lishuang Sun, Ning Xu, Chen Qian

Dear Editors and Reviewers:

We sincerely appreciate all the detailed comments and suggestions made by the reviewers. We have reproduced your comments so they can be considered point by point with our responses. We have also numbered the comments to help us organize our responses. All comments are considered carefully and summarized as follows, and all the revisions are marked in Blue color in the original paper for reviewers’ readability. Per your communication, the revised manuscript is submitted as such and we thank you for considering our manuscript for publication. The main corrections in the paper and the responds to the reviewer’s comments are as following:

Reviewer 2

1.Maybe shortly explain what is "sponge city"

Response:

Thank you for this valuable comment. The lack of explanation to the "Sponge city" may make some readers confused. According to your opinion, we briefly explain the "Sponge city" in the Introduction. (Line 40-44)

2.Explain all the abbreviations, even if some of them seem obvious.

Response:

We are very grateful for your valuable instructive advice. Many abbreviations appeared in the previous manuscript, which may cause reading difficulties for readers who are not familiar with them. In the revised manuscript, we have explained all abbreviations that first appeared in the manuscript.

3.References are not alphabetically written.

Response:

Thank you for this valuable comment. We are very sorry that we are unable to modify the issue you raised, as the format of the journal requires that references must be numbered in order of appearance in the text.

4.Line 360 explain which channels - give a few examples.

Response:

We are very grateful for your valuable instructive advice. According to our interview data, we listed two main channels of the information in "life-saving document" during the Zhengzhou extreme precipitation in this part. (Line 394-395)

5.From Table 1 or in general it is not clear the size of the sample - is it representative? Maybe I miss even more input from the actual answers from the interviews and their "interpretation".

Response:

Thank you for this valuable comment. Firstly, our interviewees came from multiple groups such as students, company employees, and government officials. We validated our research proposal through different sources of information before drew accurate conclusions. Secondly, the logic of case analysis and statistical analysis is different. Statistical analysis infers the overall characteristics through the characteristics of the sample, so it is necessary to require the sample to be representative. However, the case does not correspond to the exact overall situation, so the representativeness of the case is not the goal pursued by the case study. Nevertheless, we still believe that your suggestion are very valuable. Robert said that the typicality of a case can improve its external validity (Yin, 1994). In order to better promote the case in this article to other situations, we have added a description of the typicality of the selected case in the Methodology part. (Line 293-300)

6.There are grammar mistakes in lines: 82, 98, 172, 180, 216, 270, 286, 335, 357, 359, 463. 

Response:

Thank you for carefully reviewing the manuscript. We apologize for this error and have corrected it. We have also checked the manuscript for grammar mistakes and made modifications accordingly.

Reviewer 3 Report

Your manuscript was interesting on a personal level as my central California mountain living area was recently subject to a rainfall and flood disasters. Continuous rainfall on the order of 4-5 m over two months followed later by forest fires caused by a dry lightning event created months of havoc for area residents. Many houses not destroyed by flooding were subsequently destroyed by forest fires- some 900 houses were destroyed in one county alone with damage still being addressed some six months later. I offer some comments that recognize what you are trying to achieve to create a flood defense network and the problems involved in doing so based upon my personal experiences with US officialdom. In regard to your manuscript, the US experience involved personnel (including myself) evacuated on short notice to safe shelter areas for several weeks. No resident volunteers were requested to aid restoration efforts and many fire department resources from adjoining counties and adjoining states brought in to help fire and flood restoration efforts.  A county TV channel was set up with fire department oversight officials continuously informing evacuated residents in fire suppression progress. Basically, the threat was so great to survival that only a massive collection of county and state fire department resources, plus aircraft and helicopters dropping fire retardant liquids over two months' time, had any chance to restore normal life once again. Volunteers were not required as only fire and flood professionals were required to combat fire and flood challenges- this to eliminate causalities from well-meaning resident non-professionals unfamiliar with dangerous, life-threatening environments. While your manuscript reflects efforts to coordinate local volunteers and government officialdom to fight flood emergencies and mentions the failed 'sponge' effort, my view is that only massive government modern resources can solve large scale disaster problems and that volunteer efforts, while noble in intent, are impossible to substantially help due to lack of professional training in flood and firefighting skills. Nonetheless, your manuscript recognizes this problem and perhaps offers a first step toward getting government to put significant modern resources to work to ensure the safety of its population. For this, your manuscript is a worthy first step in the right direction!

Author Response

Dear Editors and Reviewers:

We sincerely appreciate all the detailed comments and suggestions made by the reviewers. We have reproduced your comments so they can be considered point by point with our responses. We have also numbered the comments to help us organize our responses. All comments are considered carefully and summarized as follows, and all the revisions are marked in Blue color in the original paper for reviewers’ readability. Per your communication, the revised manuscript is submitted as such and we thank you for considering our manuscript for publication. The main corrections in the paper and the responds to the reviewer’s comments are as following:

Reviewer 3

  1. Your manuscript was interesting on a personal level as my central California mountain living area was recently subject to a rainfall and flood disasters. Continuous rainfall on the order of 4-5 m over two months followed later by forest fires caused by a dry lightning event created months of havoc for area residents. Many houses not destroyed by flooding were subsequently destroyed by forest fires some 900 houses were destroyed in one county alone with damage still being addressed some six months later. I offer some comments that recognize what you are trying to achieve to create a flood defense network and the problems involved in doing so based upon my personal experiences with US officialdom. In regard to your manuscript, the US experience involved personnel (including myself) evacuated on short notice to safe shelter areas for several weeks. No resident volunteers were requested to aid restoration efforts and many fire department resources from adjoining counties and adjoining states brought in to help fire and flood restoration efforts. A county TV channel was set up with fire department oversight officials continuously informing evacuated residents in fire suppression progress. Basically, the threat was so great to survival that only a massive collection of county and state fire department resources, plus aircraft and helicopters dropping fire retardant liquids over two months' time, had any chance to restore normal life once again. Volunteers were not required as only fire and flood professionals were required to combat fire and flood challenges- this to eliminate causalities from well-meaning resident non-professionals unfamiliar with dangerous, life-threatening environments. While your manuscript reflects efforts to coordinate local volunteers and government officialdom to fight flood emergencies and mentions the failed 'sponge' effort, my view is that only massive government modern resources can solve large scale disaster problems and that volunteer efforts, while noble in intent, are impossible to substantially help due to lack of professional training in flood and firefighting skills. Nonetheless, your manuscript recognizes this problem and perhaps offers a first step toward getting government to put significant modern resources to work to ensure the safety of its population. For this, your manuscript is a worthy first step in the right direction!

Response:

Thank you for your valuable comment. We have made modifications to the Discussion based on your comment. We fully agree with your viewpoint that in the face of severe disasters, the government is undoubtedly the most critical force, and spontaneous volunteers cannot become the leader in emergency response limited by scale and skills. Your personal experience has deepened our understanding of this issue. In the Zhengzhou extreme precipitation, more than 3000 people were saved through the "life-saving document", which shows us the potential of public participation and the possibility of Tech for Good. Thus, spontaneous volunteering can undoubtedly become an important helper for official response in the future. At the same time, in order to avoid the dilemma of “noble intentions but do a disservice”, we have proposed three countermeasures to respond to your comments and added references to support our discussion. Thank you again for your suggestion, which prompted us to think and improve our manuscript. (Line 518-547)

1. Lodree Jr, E.J.; Davis, L.B. Empirical analysis of volunteer convergence following the 2011 tornado disaster in Tuscaloosa, Alabama. Natural Hazards 2016, 84, 1109-1135.

2. Boersma, K.; Kraiukhina, A.; Larruina, R.; Lehota, Z.; Nury, E.O. A port in a storm: Spontaneous volunteering and grassroots movements in Amsterdam. A resilient approach to the (European) refugee crisis. Soc Policy Admin 2019, 53, 728-742.

3. Lai, T.; Wang, W.Q. Attribution of community emergency volunteer behaviour during the covid-19 pandemic: A study of community residents in Shanghai, China. Voluntas 2023, 34, 239-251.

4. Zhu, Y.Y.; Zhuang, J.; Liu, B.H.; Liu, H.; Ren, J.J.; Zhao, M.M. The moderating effect of covid-19 risk perception on the relationship between empathy and covid-19 volunteer behavior: A cross-sectional study in Jiangsu, China. Front Public Health 2022, 10, 863613.

5. Jiang, H.; Zhang, Y.; Guo, W.D.; Cheng, W.; Peng, J. Online collaborative documents as media logic: The mediatization of risk response in the post-pandemic era. Front Psychol 2022, 13, 892569.